

# lab: an R package for generating analysis-ready data from laboratory records

Yi-Ju Tseng[1,2], Chun Ju Chen[3] and Chia Wei Chang[1]

[1] Department of Computer Science, National Yang Ming Chiao Tung University, Hsinchu, Taiwan
[2] Computational Health Informatics Program, Boston Children's Hospital, Boston, MA, United States of America
[3] Department of Information Management, National Taiwan University, Taipei, Taiwan

## ABSTRACT

**Background:** Electronic health records (EHRs) play a crucial role in healthcare decision-making by giving physicians insights into disease progression and suitable treatment options. Within EHRs, laboratory test results are frequently utilized for predicting disease progression. However, processing laboratory test results often poses challenges due to variations in units and formats. In addition, leveraging the temporal information in EHRs can improve outcomes, prognoses, and diagnosis predication. Nevertheless, the irregular frequency of the data in these records necessitates data preprocessing, which can add complexity to time-series analyses.

**Methods:** To address these challenges, we developed an open-source R package that facilitates the extraction of temporal information from laboratory records. The proposed *lab* package generates analysis-ready time series data by segmenting the data into time-series windows and imputing missing values. Moreover, users can map local laboratory codes to the Logical Observation Identifier Names and Codes (LOINC), an international standard. This mapping allows users to incorporate additional information, such as reference ranges and related diseases. Moreover, the reference ranges provided by LOINC enable us to categorize results into normal or abnormal. Finally, the analysis-ready time series data can be further summarized using descriptive statistics and utilized to develop models using machine learning technologies.

**Results:** Using the *lab* package, we analyzed data from MIMIC-III, focusing on newborns with patent ductus arteriosus (PDA). We extracted time-series laboratory records and compared the differences in test results between patients with and without 30-day in-hospital mortality. We then identified significant variations in several laboratory test results 7 days after PDA diagnosis. Leveraging the time series–analysis-ready data, we trained a prediction model with the long short-term memory algorithm, achieving an area under the receiver operating characteristic curve of 0.83 for predicting 30-day in-hospital mortality in model training. These findings demonstrate the lab package's effectiveness in analyzing disease progression.

**Conclusions:** The proposed *lab* package simplifies and expedites the workflow involved in laboratory records extraction. This tool is particularly valuable in assisting clinical data analysts in overcoming the obstacles associated with heterogeneous and sparse laboratory records.

Corresponding author
Yi-Ju Tseng, yjtseng@nycu.edu.tw

## INTRODUCTION

Electronic health records (EHRs), which have been widely adopted in medical institutions, serve as comprehensive repositories of clinical data and offer valuable resources for clinical research (*Safran et al., 2007*; *Murdoch & Detsky, 2013*; *Rudin et al., 2020*; *Macias, Remy & Barda, 2023*). Leveraging secondary EHR data, clinicians and researchers can gain deeper insights into disease progression, identify appropriate treatments strategies (*Safran et al., 2007*; *Murdoch & Detsky, 2013*), and make data-driven healthcare decisions in healthcare (*Hersh, 2007*; *Rudin et al., 2020*). Within EHR, laboratory test results play a crucial role (*Murdoch & Detsky, 2013*; *Shickel et al., 2017*), providing valuable information for developing risk models and predicting disease progression (*Perotte et al., 2015*; *Tseng et al., 2019*; *Norgeot et al., 2019*; *Li et al., 2023*; *Chien et al., 2023*). These predictive models can be further enhanced by incorporating machine learning and deep learning technologies, enabling more accurate assessments and predictions.

Unlike diagnosis or procedure records, laboratory records present unique challenges in EHR analyses. These records include information about the tests performed and contain test results with variations in units and formats. For example, hospitals may record C-reactive protein (CRP) levels in mg/dL or mg/L. Furthermore, inconsistencies in coding systems across institutes or EHR systems make data collection and analysis more difficult, as laboratory tests are often coded using local coding systems (*Lin, Vreeman & Huff, 2011*; *Zunner et al., 2013*; *Miotto et al., 2016*). Sometimes, the same laboratory test may be coded differently, further complicating data aggregation. To address these challenges, there is a need for a tool that enables users to standardize and manipulate laboratory data using the Logical Observation Identifier Names and Codes (LOINC), which serves as the international standard terminology for laboratory and clinical observations (*Abhyankar, Demner-Fushman & McDonald, 2012*). By mapping local coding systems to LOINC, analysts can access standard names, reference ranges, and standard units for each test. Additionally, LOINC allows for searching laboratory items based on related information such as disease names, specimens, and other relevant terminology.

Analysis of EHRs incorporating temporal information has been shown to improve outcomes, prognoses, and diagnosis predictions (*Singh et al., 2015*; *Liu et al., 2019*; *Placido et al., 2023*). Furthermore, prediction models developed using longitudinal laboratory records have improved predictive capabilities (*Perotte et al., 2015*; *Norgeot et al., 2019*). However, the inherent features of EHRs, including heterogeneity and sparseness, pose challenges in extracting and analyzing meaningful data (*Anhøj, 2003*). The irregular frequency and varying data collection intervals, especially in real-world healthcare systems, are widely recognized as problems that need to be addressed for practical analysis (*Yadav et al., 2018*). Sparseness and missingness in EHRs can arise for various reasons, including instances when certain tests or measurements were not conducted, data was not consistently recorded or documented, or information was omitted or unavailable for specific patients or encounters (*Holmes et al., 2021*; *Knevel & Liao, 2023*). One approach to

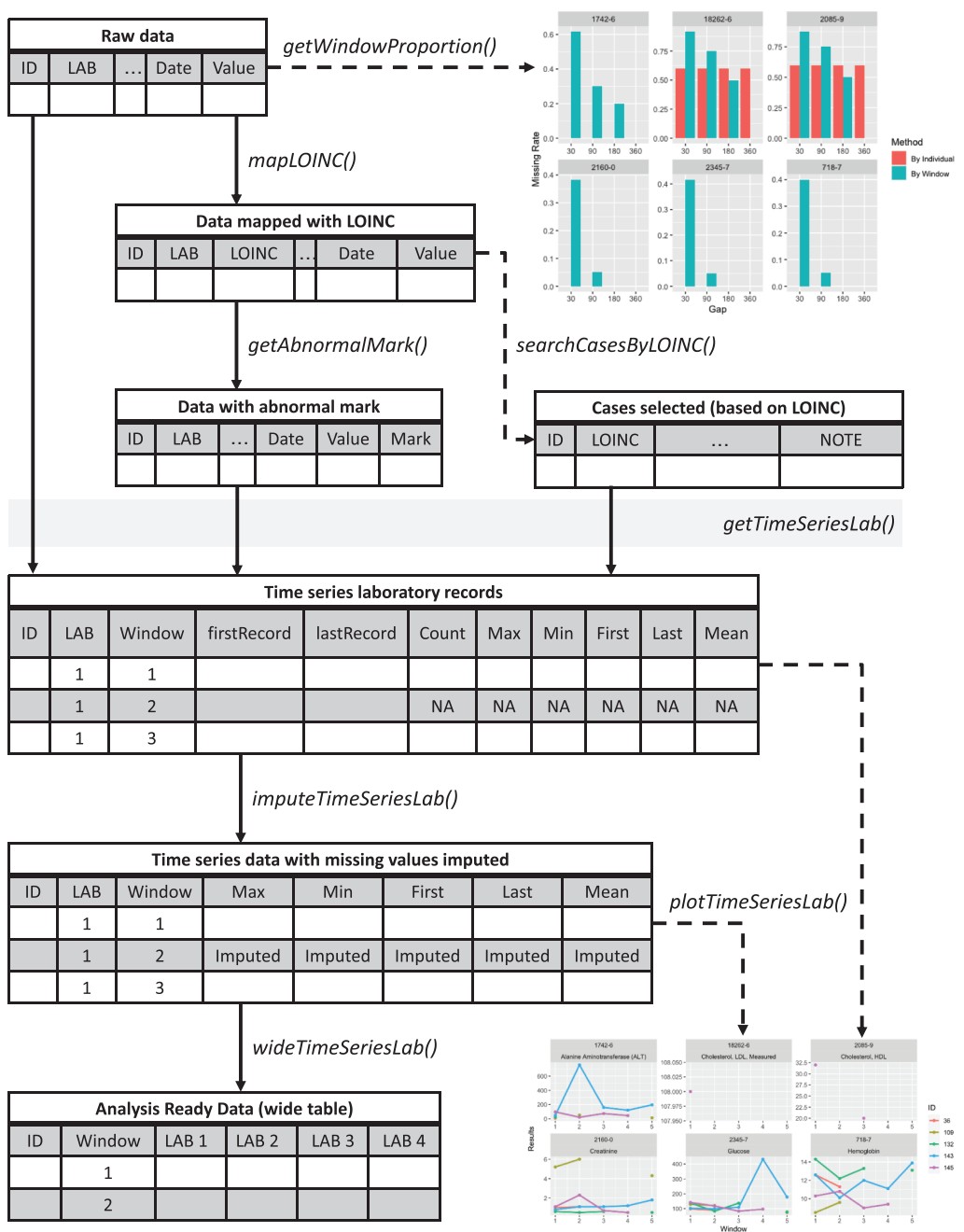

**Figure 1 Overview of the *lab* package.** Starting from the raw data, users can explore the data completeness of each length of time window with getWindowProportion(), map data to LOINC standard with mapLOINC(), and identify laboratory test results as normal or abnormal with getAnnormalMark(). Combining the laboratory test results with normal or abnormal flags and the selected time window length, users can generate time series laboratory records with getTimeSeriesLab(). After imputation with imputeTimeSeriesLab(), users can generate analysis-ready data with wideTimeSeriesLab() and visualize the trends of laboratory test results with plotTimeSeriesLab().

address data irregularity involves developing algorithms capable of handling irregularly sampled time-series data with missing values (*Horn et al., 2020*; *Wang et al., 2021*). Another approach is the design of data preprocessing methods, such as resampling and imputation, to aggregate data into a regularly-sampled format before utilizing it in commonly used time-series machine learning algorithms (*Tseng et al., 2015*). These algorithms typically rely on regular time intervals, such as autoregressive integrated moving averages and basic recurrent neural network models.

Therefore, we developed an open-source R package *lab* that facilitates the retrieval of detailed information from laboratory records. This package includes functionalities such as mapping local laboratory codes to the international standard, annotating abnormal results, segmenting data into time-series windows, summarizing data using descriptive statistics, imputing missing values, and generating analysis-ready datasets. By utilizing the *lab* package, analysts can streamline and expedite the workflow of laboratory report extraction. Moreover, it can assist clinical data analysts in producing concise and reproducible research methodology, thereby enhancing collaboration and knowledge sharing within the field.

## MATERIALS AND METHODS

An overview of the *lab* package is presented in Fig. 1. The sample codes described below are available at https://github.com/DHLab-TSENG/lab-paper/blob/main/SampleCode.md. The development version of the package can be found on GitHub (https://github.com/DHLab-TSENG/lab) and is accessible through the *remotes* R package, which enables installation of packages from GitHub (*Csárdi et al., 2021*).

To install the *lab* R package, users can enter the following commands in an R session:

```
# install.packages("remotes")
remotes::install_github("DHLab-TSENG/lab")
library(lab)
```

### Data input

Raw, original laboratory records consist of at least four components: patient identification numbers, dates of tests performed or reported, laboratory item codes for identifying specific laboratory tests, and test results presented in either numerical or categorical formats. While LOINC codes are international standards for laboratory test identification, local coding systems are commonly utilized. It is worth noting that some laboratory coding systems may assign a single laboratory item code to multiple tests (*Liu et al., 2007*). Therefore, in *lab* package, the laboratory item codes can be listed in multiple columns, allowing for the specification of an analyte, specimen type, and other critical information required for laboratory test identification. All the selected columns serve as the primary key, ensuring the identification of a unique laboratory test and preventing ambiguity.

The *lab* package includes sample data called `labSample`, which was generated and modified from Medical Information Mart for Intensive Care-III (MIMIC-III) (*Johnson et al., 2016*). This sample data comprises 1,744 laboratory test results for seven laboratory items given to five patients. The patient IDs are in the `SUBJECT_ID` column, the dates that the tests were performed or reported are in the `CHARTTIME` column, the laboratory item codes for test identification are in the `ITEMID` column, and the numeric or categorical test results are in the `VALUENUM` column.

```
head(labSample)
#>      SUBJECT_ID    ITEMID    CHARTTIME    VALUENUM    VALUEUOM       FLAG
#> 1:           36     50811   2131-05-18        12.7        g/dL   abnormal
#> 2:           36     50912   2131-05-18         1.2       mg/dL
#> 3:           36     51222   2131-05-18        11.9        g/dL   abnormal
#> 4:           36     50912   2131-05-19         1.3       mg/dL   abnormal
#> 5:           36     50931   2131-05-19       160.0       mg/dL   abnormal
#> 6:           36     51222   2131-05-19         9.6        g/dL   abnormal
```

## LOINC mapping

LOINC provides information on test names (labels), specimens, and test categories for certain laboratory tests, along with other details such as reference ranges and related names. These parameters are used by the *lab* package to wrangle laboratory records in subsequent steps. Therefore, if LOINC is not the default coding system, it is recommended to map local laboratory codes to LOINCs using the `mapLOINC` function. Users have to provide a user-defined mapping table that includes the local codes and their corresponding LOINCs. The `mapLOINC` function directly appends LOINCs to the original table when the local codes match. If there is no match, the field will be filled with "NA" to indicate the absence of a corresponding LOINC code. To demonstrate the `mapLOINC` function, we included a sampling mapping table called `mapSample`, provided by MIMIC III.

```
head(mapSample)
#>      ITEMID                            LABEL    FLUID     CATEGORY      LOINC
#> 1:    50811                       Hemoglobin    Blood    Blood Gas      718-7
#> 2:    50861    Alanine Aminotransferase (ALT)   Blood    Chemistry     1742-6
#> 3:    50904                 Cholesterol, HDL    Blood    Chemistry     2085-9
#> 4:    50906        Cholesterol, LDL, Measured   Blood    Chemistry    18262-6
#> 5:    50912                       Creatinine    Blood    Chemistry     2160-0
#> 6:    50931                          Glucose    Blood    Chemistry     2345-7
```

```
loincSample <- mapLOINC(labData = labSample, labItemColName = ITEMID, mappingTable =
mapSample)
loincSample[loincSample$SUBJECT_ID==36&loincSample$CHARTTIME=="2131-05-17",]
#>      ITEMID   SUBJECT_ID    CHARTTIME   VALUENUM    VALUEUOM      FLAG
#> 1:    50811          36   2131-05-17       11.7        g/dL   abnormal
#> 2:    50861          36   2131-05-17       12.0        IU/L
#> 3:    50912          36   2131-05-17        2.0       mg/dL   abnormal
#> 4:    50931          36   2131-05-17      125.0       mg/dL   abnormal
#> 5:    51222          36   2131-05-17        7.1        g/dL   abnormal
#>                                   LABEL     FLUID    CATEGORY     LOINC
#> 1:                            Hemoglobin     Blood   Blood Gas     718-7
#> 2:       Alanine Aminotransferase (ALT)     Blood   Chemistry    1742-6
#> 3:                            Creatinine     Blood   Chemistry    2160-0
#> 4:                               Glucose     Blood   Chemistry    2345-7
#> 5:                            Hemoglobin     Blood  Hematology     718-7
```

After users map local laboratory item codes with LOINC, the sex-sensitive range information provided by LOINC can be used to annotate abnormal results through the `getAbnormalMark` function. The results are annotated as abnormal when the values deviate from a specific test's expected or normal range. It primarily relates to the interpretation of test results within a clinical context. In addition, LOINC provides relevant terms, such as common names of a laboratory test or disease, which can be used to search for specific laboratory test codes through the `searchCasesByLOINC` function.

```
loincMarkedSample <- getAbnormalMark(labData = loincSample,
                            idColName = SUBJECT_ID,
                            labItemColName = LOINC,
                            valueColName = VALUENUM,
                            genderColName = GENDER,
                            genderTable = patientSample,
                            referenceTable = refLOINC)
head(loincMarkedSample)
#>      ITEMID     ID     CHARTTIME   Value    VALUEUOM         FLAG
#> 1:    50861     36    2131-04-30       8        IU/L
#> 2:    50861     36    2131-05-17      12        IU/L
#> 3:    50861     36    2134-05-14      12        IU/L
#> 4:    50861    109    2138-07-03      14        IU/L
#> 5:    50861    109    2142-03-21      46        IU/L     abnormal
#> 6:    50861    109    2142-01-09      10        IU/L
#>                               LABEL   FLUID   CATEGORY     LOINC   ABMark
#> 1:   Alanine Aminotransferase (ALT)   Blood  Chemistry    1742-6     <NA>
#> 2:   Alanine Aminotransferase (ALT)   Blood  Chemistry    1742-6     <NA>
#> 3:   Alanine Aminotransferase (ALT)   Blood  Chemistry    1742-6     <NA>
#> 4:   Alanine Aminotransferase (ALT)   Blood  Chemistry    1742-6     <NA>
#> 5:   Alanine Aminotransferase (ALT)   Blood  Chemistry    1742-6        H
#> 6:   Alanine Aminotransferase (ALT)   Blood  Chemistry    1742-6     <NA>
```

```
caseCreatinine <- searchCasesByLOINC(labData = loincSample,
                                     idColName = SUBJECT_ID,
                                     loincColName = LOINC,
                                     dateColName = CHARTTIME,
                                     condition = "Creatinine",
                                     isSummary = TRUE)

head(caseCreatinine)
#>           ID       LOINC       Count      firstRecord      lastRecode
#> 1:        36      2160-0         37       2131-04-30       2134-05-20
#> 2:       109      2160-0        238       2137-11-04       2142-08-30
#> 3:       132      2160-0         32       2115-05-06       2116-04-08
#> 4:       143      2160-0         60       2154-12-25       2155-10-22
#> 5:       145      2160-0        162       2144-03-29       2145-02-22
```

Because reference ranges can differ between institutions (*Smalls & Fischbach, 1982*), the *lab* package only includes reference values suggested by LOINC. However, users can also import a mapping table containing self-defined reference ranges for each laboratory test item. These reference ranges can be specified separately for males and females if needed. Once mapping with the reference range table is completed, values falling outside the designated range are marked as abnormal, with L representing "Low" and H representing "High" abnormality. This marking system aids in visualization and facilitates further analysis.

## Time-series analysis

The *lab* package provides users with the capability to segment laboratory test results into multiple, consecutive non-overlapping time windows. The index date for the time windows can be defined as the first or last record date for each individual, or a specific date, such as the first diagnosis date for a particular disease. To assist users in determining an appropriate window size (*e.g.*, 30, 90, or 180 days), the *lab* package includes a plot function called `plotWindowProportion`. This function allows the visualization of the distribution of missing values proportions after segmenting the data. Users can decide on the proper window size for data segmentation by observing and comparing missing rates of each time period. The plot also depicts the frequency of patients undergoing laboratory tests (Fig. 2). The function offers two approaches for missing rates calculation: "By Individuals," which displays the proportion of patients who never received the laboratory
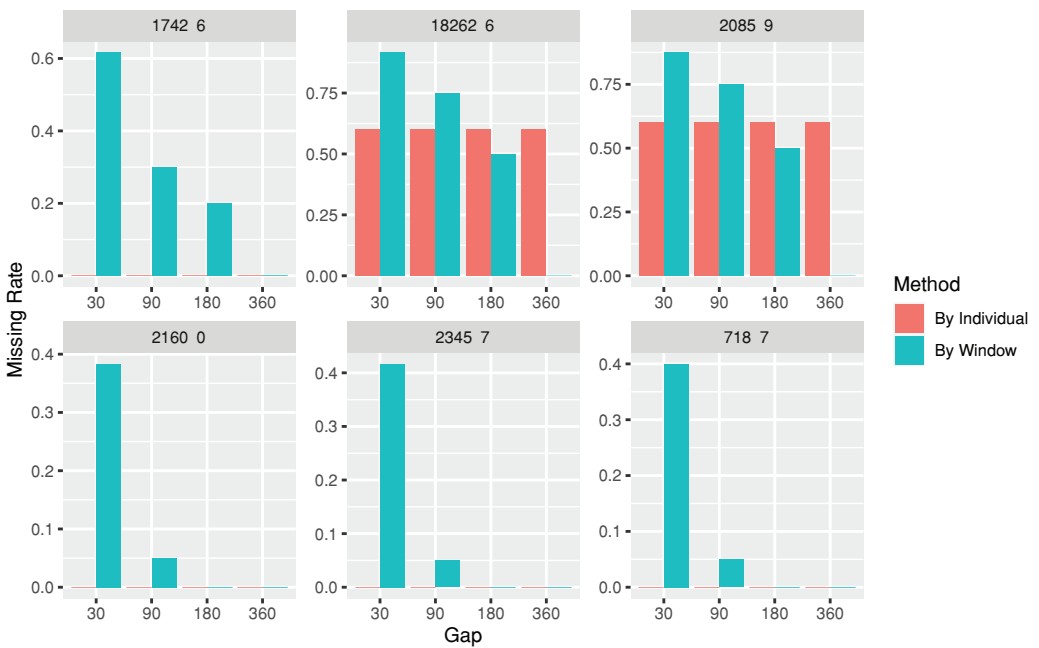

**Figure 2 Missing rate of laboratory records by individuals or by time windows.** The subgraphs are the missing rate of each laboratory test, with the identification code on the upper banner. The "By Individual" missing rate is the proportion of patients who never received the laboratory test. The "By Window" missing rate is defined by the proportion of windows without laboratory test results.

```
windowProportion <- plotWindowProportion(labData = loincSample,
                                         idColName = SUBJECT_ID,
                                         labItemColName = LOINC,
                                         dateColName = CHARTTIME,
                                         indexDate = first,
                                         gapDate = c(30, 90, 180, 360),
                                         studyPeriodStartDays=0,
                                         studyPeriodEndDays=360)
head(windowProportion$missingData)
#>                  LAB         Gap              Method         Proportion
#> 1:            1742-6          30        By Individual                 0
#> 2:            1742-6          30        By Individual                 0
#> 3:            1742-6          30        By Individual                 0
#> 4:            1742-6          30        By Individual                 0
#> 5:            1742-6          30        By Individual                 0
#> 6:            2160-0          30        By Individual                 0
print(windowProportion$graph)
```

test, and "By Window," which displays the proportion of windows without laboratory test results.

After the index date and window size are determined (*e.g.*, the first test record date is set as the index date, and 30-days is set as the window size), the descriptive statistics, including

the total number of tests within a window, maximum test result, minimum test result, test result mean, and result nearest to the index date, can be obtained using the

```
timeSeriesData <- getTimeSeriesLab(labData = loincSample,
                                   idColName = SUBJECT_ID,
                                   labItemColName = LOINC + LABEL,
                                   dateColName = CHARTTIME,
                                   valueColName = VALUENUM,
                                   indexDate = first,
                                   gapDate = 30,
                                   completeWindows = TRUE)
head(timeSeriesData)
#>      ID    LOINC                            LABEL   Window   Count   Max   Min   Mean   Nearest
#> 1:   36   1742-6   Alanine Aminotransferase (ALT)        1       2    12     8     10         8
#> 2:   36   1742-6   Alanine Aminotransferase (ALT)        2      NA    NA    NA     NA        NA
#> 3:   36   1742-6   Alanine Aminotransferase (ALT)        3      NA    NA    NA     NA        NA
#> 4:   36   1742-6   Alanine Aminotransferase (ALT)        4      NA    NA    NA     NA        NA
#> 5:   36   1742-6   Alanine Aminotransferase (ALT)        5      NA    NA    NA     NA        NA
#> 6:   36   1742-6   Alanine Aminotransferase (ALT)        6      NA    NA    NA     NA        NA
#>         firstRecord                     lastRecode
#> 1:      2131-04-30                     2131-05-17
#> 2:            <NA>                           <NA>
#> 3:            <NA>                           <NA>
#> 4:            <NA>                           <NA>
#> 5:            <NA>                           <NA>
#> 6:            <NA>                           <NA>
```

`getTimeSeriesLab` function. The time window series is a complete sequence in order by default, even if no records are available for a certain window.

A function for plotting line charts, `plotTimeSeriesLab`, is available to visualize long-

```
timeSeriesPlot <- plotTimeSeriesLab(labData = timeSeriesData,
                                    idColName = ID,
                                    labItemColName = LOINC + LABEL,
                                    timeMarkColName = Window,
                                    valueColName = Nearest,
                                    timeStart = 1,
                                    timeEnd  = 5,
                                    abnormalMarkColName = NULL)
plot(timeSeriesPlot)
```

term trends in the test results (Fig. 3). "L" (Low) and "H" (High) icons appear in the legends if abnormal values are identified.
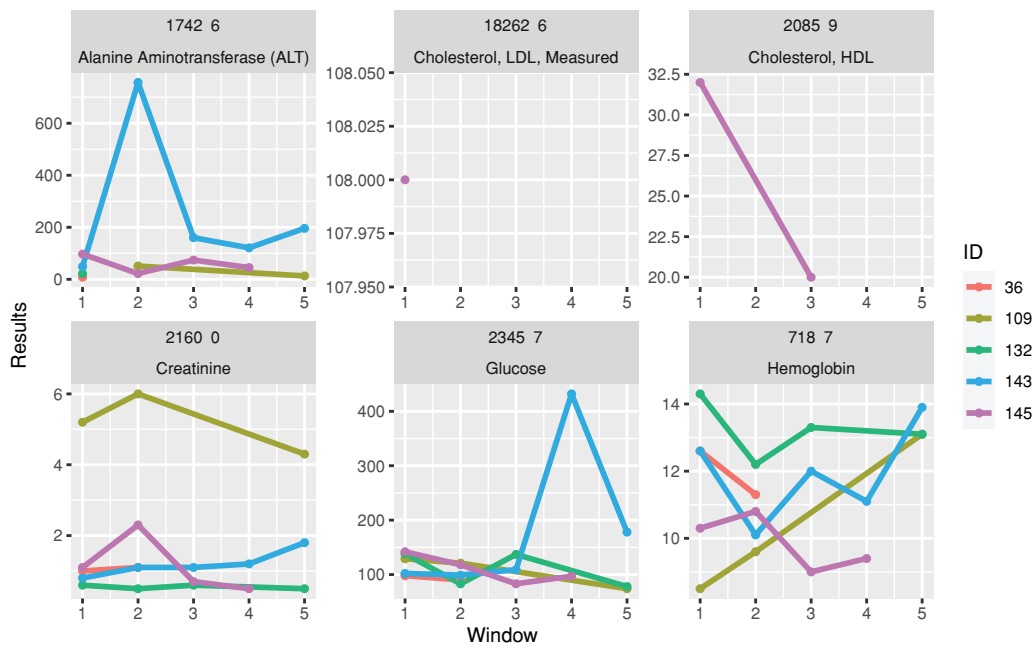

**Figure 3 Trends of laboratory records.** The subgraphs are the trends of each laboratory result, with the identification code and test name on the upper banners. The color stands for laboratory results from different individuals.

## Data imputation

After completing time-series processing, the imputation function `imputeTimeSeriesLab` can be applied to impute missing data. Certain modeling algorithms cannot handle missing values in the dataset and necessitate complete data without any missing values for calculations or modeling. Therefore, it is essential to impute missing values before applying these algorithms. The imputation methods implemented in the function include simple mean imputation, interpolation, and "next observation carry forward," which are frequently used in practice (*Jakobsen et al., 2017*). Simple mean imputation replaces missing values in a dataset by substituting them with the mean value of the available data for that specific lab test variable. Interpolation replaces missing values by interpolating values based on the temporal trends observed in the data. Next

```
timeSeriesData[timeSeriesData$ID==36&timeSeriesData$LOINC=="2160-0"]
#>      ID    LOINC      LABEL  Window   Count  Max       Min       Mean  Nearest  firstRecord
#> 1:   36  2160-0  Creatinine       1      23  2.0       0.7  1.2347826      1.0  2131-04-30
#> 2:   36  2160-0  Creatinine       2       1  1.1       1.1  1.1000000      1.1  2131-06-22
#> 3:   36  2160-0  Creatinine       3      NA   NA        NA         NA       NA       <NA>
#> 4:   36  2160-0  Creatinine       4      NA   NA        NA         NA       NA       <NA>
#> 5:   36  2160-0  Creatinine       5      NA   NA        NA         NA       NA       <NA>
#> 6:   36  2160-0  Creatinine       6      NA   NA        NA         NA       NA       <NA>
#> 7:   36  2160-0  Creatinine       7      NA   NA        NA         NA       NA       <NA>
#> 8:   36  2160-0  Creatinine       8      NA   NA        NA         NA       NA       <NA>
#> 9:   36  2160-0  Creatinine       9       2  1.2       1.2  1.2000000      1.2  2132-01-09
```

```
fullTimeSeriesData <- imputeTimeSeriesLab(labData = timeSeriesData,
                                          idColName = ID,
                                          labItemColName = LOINC + LABEL,
                                          windowColName = Window,
                                          valueColName = Mean & Nearest,
                                          impMethod = NOCB,
                                          imputeOverallMean = FALSE)
fullTimeSeriesData[timeSeriesData$ID==36&timeSeriesData$LOINC=="2160-0"]
#>          ID      LOINC      LABEL   Window      Mean    Nearest      imputed
#> 1:       36     2160-0   Creatinine      1   1.2347826       1.0        FALSE
#> 2:       36     2160-0   Creatinine      2   1.1000000       1.1        FALSE
#> 3:       36     2160-0   Creatinine      3   1.1000000       1.1         TRUE
#> 4:       36     2160-0   Creatinine      4   1.1000000       1.1         TRUE
#> 5:       36     2160-0   Creatinine      5   1.1000000       1.1         TRUE
#> 6:       36     2160-0   Creatinine      6   1.1000000       1.1         TRUE
#> 7:       36     2160-0   Creatinine      7   1.1000000       1.1         TRUE
#> 8:       36     2160-0   Creatinine      8   1.1000000       1.1         TRUE
#> 9:       36     2160-0   Creatinine      9   1.2000000       1.2        FALSE
```

observation carry forward imputation involves propagating the last observed value to fill in the missing field until the next observed value is encountered. In cases where interpolation is not applicable (*e.g.*, missing in the first window), a simple mean imputation method is

```
wideTimeSeriesData <- wideTimeSeriesLab(labData = fullTimeSeriesData,
                                        idColName = ID,
                                        labItemColName = LOINC + LABEL,
                                        windowColName = Window,
                                        valueColName = Nearest)
head(wideTimeSeriesData)
#>             ID        Window   1742-6_Alanine Aminotransferase (ALT)
#> 1:          36             1                                       8
#> 2:          36             2                                      10
#> 3:          36             3                                      10
#> 4:          36             4                                      10
#> 5:          36             5                                      10
#> 6:          36             6                                      10
#> 18262-6_Cholesterol, LDL, Measured 2085-9_Cholesterol, HDL 2160-0_Creatinine
#> 1:                                NA                      NA              1.00
#> 2:                                NA                      NA              1.10
#> 3:                                NA                      NA              1.05
#> 4:                                NA                      NA              1.05
#> 5:                                NA                      NA              1.05
#> 6:                                NA                      NA              1.05
```

```
#> 2345-7_Glucose 718-7_Hemoglobin
#> 1:            98.0000              12.60000
#> 2:            90.0000              11.30000
#> 3:           102.1667              12.83333
#> 4:           102.1667              12.83333
#> 5:           102.1667              12.83333
#> 6:           102.1667              12.83333
```

used. Furthermore, if an individual has never undergone a particular test, the mean of the test results from all individuals in the dataset can be imputed.

## Analysis-ready data generation

The `wideTimeSeriesLab` function is designed to convert longitudinal data into a wide format, thereby generating analysis-ready data. This transformation converts imputed and non-imputed data into time series analysis-ready format.

```
knnImputedData <- imputeKNN(labData = wideTimeSeriesData,
                            idColName = ID + Window,
                            k = 1)
head(knnImputedData)
#>            ID              Window   1742-6_Alanine Aminotransferase (ALT)
#> 1:         36                   1                                      8
#> 2:         36                   2                                      8
#> 3:         36                   3                                      8
#> 4:         36                   4                                      8
#> 5:         36                   5                                      8
#> 6:         36                   6                                      8
#> 18262-6_Cholesterol, LDL, Measured 2085-9_Cholesterol, HDL 2160-0_Creatinine
#> 1:                                108                      32                1.0
#> 2:                                108                      32                1.1
#> 3:                                108                      32                1.1
#> 4:                                108                      32                1.1
#> 5:                                108                      32                1.1
#> 6:                                108                      32                1.1
#> 2345-7_Glucose 718-7_Hemoglobin
#> 1:            98              12.6
#> 2:            90              11.3
#> 3:            90              11.3
#> 4:            90              11.3
#> 5:            90              11.3
#> 6:            90              11.3
```

**Table 1 Functions in the *lab* package.**

| Function name | Purpose |
| --- | --- |
| Data Preprocessing | |
| mapLOINC | Maps local laboratory codes with LOINC |
| getAbnormalMark | Annotates abnormalities of laboratory test results based on the reference range |
| Data Selection | |
| searchCasesByLOINC | Searches cases using keywords provided by LOINC |
| Data Transformation | |
| getTimeSeriesLab | Obtains time-series data with moving windows |
| imputeTimeSeriesLab | Imputes missing values in time-series data |
| wideTimeSeriesLab | Transforms time-series data into an analysis-ready format |
| imputeKNN | Imputes missing values in time-series data with analysis-ready format using the k nearest neighbors approach |
| Visualization | |
| plotWindowProportion | Plots data missing rates in each time-series window |
| plotTimeSeriesLab | Plots the trends of laboratory results over time |

### Data imputation—*k* nearest neighbors approach

In addition to the aforementioned basic data imputation approaches, we provide an alternative method for imputing missing values using the *k* nearest neighbors (kNN) algorithm based on similar cases. Unlike the basic data imputation approaches, this method should be applied to the analysis-ready wide format dataset. The *lab* package provides imputeKNN function, which allows users to impute missing values by leveraging their *k* nearest neighbors (*Troyanskaya et al., 2001*).

### Summary

A list of the functions in the *lab* package is presented in Table 1.

## RESULTS

To illustrate the lab package's main functionalities and common workflow, we analyze data from MIMIC-III on newborns diagnosed with patent ductus arteriosus (PDA) (*Johnson et al., 2016*). MIMIC-III is a publicly available database that contains de-identified health-related data from approximately 60,000 patients admitted to the critical care unit of the Beth Israel Deaconess Medical Center between 2001 and 2012.

We first identified cases of PDA using the *dxpr* package (*Tseng, Chiu & Chen, 2021*), a package developed by our team for diagnosis and procedure data analysis. We defined patients with PDA as those with at least one PDA diagnosis (ICD-9-CM 7470*). The process used to identify patients with PDA is described in a previous study (*Tseng, Chiu & Chen, 2021*). We defined the primary outcome of interest as 30-day in-hospital mortality. The original laboratory records obtained from MIMIC-III were used to prepare time-series laboratory records and examine the differences in the laboratory test results of patients with and without 30-day in-hospital mortality using the *lab* package.

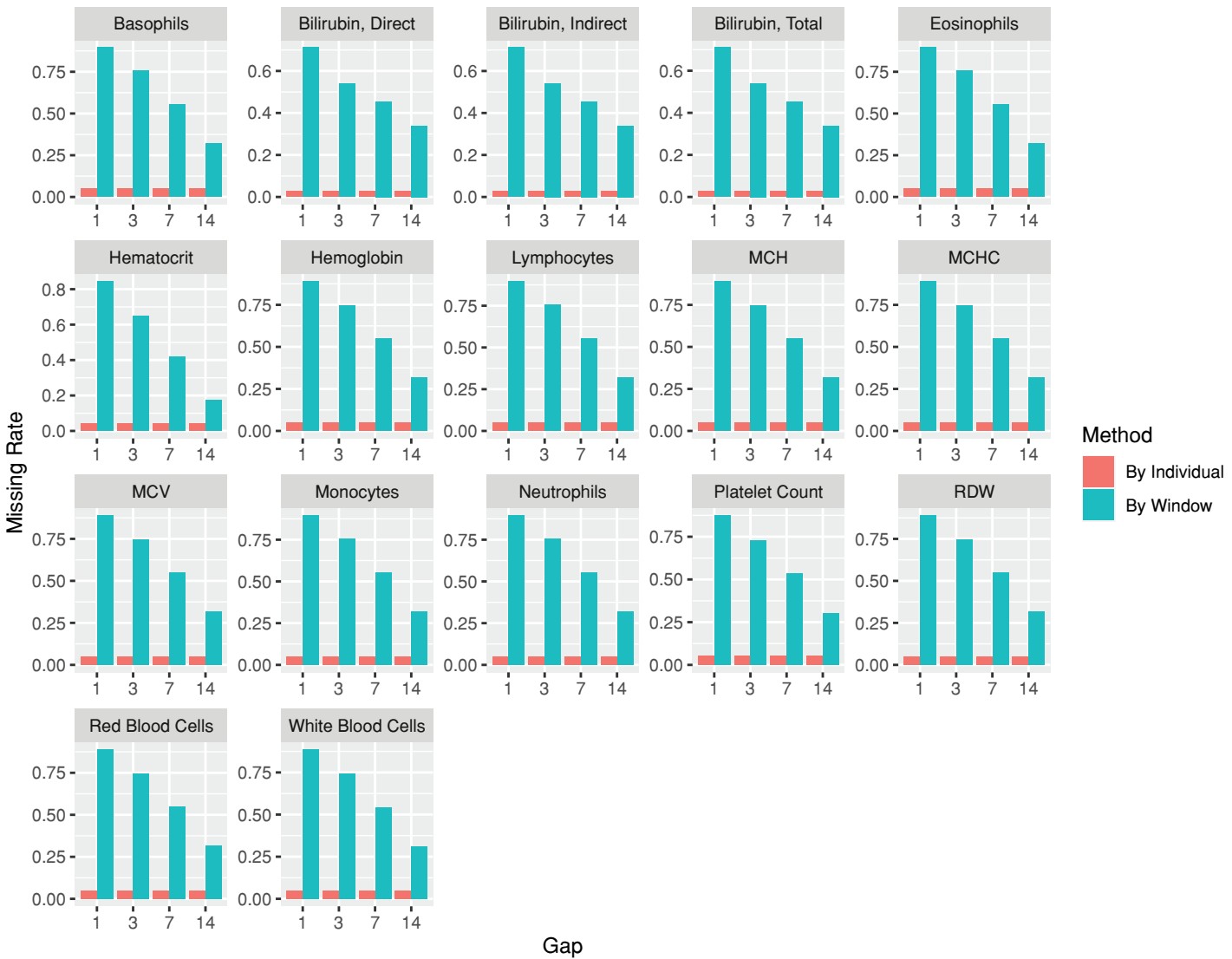

**Figure 4 Missing rate of laboratory test results from patients with patent ductus arteriosus, as determined by individuals or by time windows.** The subgraphs are the missing rate of each laboratory test, with the test name on the upper banner.

The codes used for the use case are provided at https://github.com/DHLab-TSENG/lab-paper/blob/main/UseCase.md. After we identified patients with PDA from a cohort of newborns whose data were obtained from MIMIC-III ($n$ = 7,833) by using the *dxpr* package, we included data from 381 patients with PDA, among whom 12 (3.1%) experienced in-hospital mortality within 30 days. To compare the laboratory results of the patients with and without 30-day in-hospital mortality, we included the laboratory tests administered to more than 95% of the patients. The index date was the date of PDA diagnosis. The `plotWindowProportion` function was employed to visualize the proportion of missing values in different time windows, aiding in determining an appropriate window size for segmenting the laboratory records. Figure 4 illustrates that

**Table 2 Laboratory test results of patients with patent ductus arteriosus.**

| Laboratory results (mean (SD)) | Without 30-day in-hospital mortality | With 30-day in-hospital mortality | P value |
|---|---|---|---|
| Patient, n | 367 | 12 | |
| Total bilirubin | 4.87 (2.50) | 2.47 (1.21) | 0.001 |
| Hematocrit | 46.67 (6.49) | 43.68 (7.21) | 0.119 |
| Basophils | 0.23 (0.55) | 0.33 (0.65) | 0.543 |
| Eosinophils | 1.85 (2.06) | 1.42 (1.62) | 0.469 |
| Hemoglobin | 15.55 (2.23) | 14.71 (2.13) | 0.197 |
| Lymphocytes | 54.68 (20.25) | 55.51 (20.16) | 0.889 |
| Monocytes | 7.55 (4.47) | 10.25 (4.37) | 0.04 |
| Neutrophils | 32.11 (18.49) | 28.80 (16.61) | 0.542 |
| Platelet count | 236.71 (82.48) | 226.50 (73.40) | 0.673 |
| Mean corpuscular hemoglobin | 37.95 (2.57) | 40.29 (1.82) | 0.002 |
| Mean corpuscular hemoglobin concentration | 33.31 (1.00) | 33.37 (1.20) | 0.855 |
| Mean corpuscular volume | 114.03 (8.05) | 121.00 (7.66) | 0.003 |
| Red blood cell volume distribution width | 17.07 (1.48) | 16.68 (1.83) | 0.372 |
| Red blood cells | 4.11 (0.60) | 3.66 (0.57) | 0.011 |
| White blood cells | 10.19 (7.15) | 7.33 (3.94) | 0.17 |

using a 1- or 3-day window could result in a large ratio of missing records, thereby affecting the analytical outcome. Using a 7-day window decreased the missing rate to approximately 50%, leading us to select 7 days as the segmenting window size. The analysis-ready data generated using the `wideTimeSeriesLab` function can be passed to the *tableone* (*Yoshida & Bartel, 2022*) package to create objects summarizing all laboratory test results stratified by the patients with and without 30-day in-hospital mortality. Statistical tests were conducted accordingly. The results revealed significant differences in five out of the 15 laboratory test results, including total bilirubin levels, monocyte levels, mean corpuscular hemoglobin levels, mean corpuscular volume, and red blood cell counts, between patients with and without 30-day in-hospital mortality at 7 days after their PDA diagnosis (The first time window; $p < 0.05$; Table 2). We applied the same process to each window to identify the key indicators of 30-day in-hospital mortality during different hospitalization periods. The time series analysis-ready data can be used as an input for training prediction models, employing time-series deep learning algorithms such as the long short-term memory algorithm with the *keras* (*Falbel, Allaire & Chollet, 2022*) package. The training performance reached an area under the receiver operating characteristic curve of 0.83 after 100 epochs with the long short-term memory algorithm.

## DISCUSSION

Utilizing the temporal information derived from time-series measurements to predict patient prognoses is a highly necessary yet challenging task. Tools that facilitate data processing for time-series analyses have emerged as the preferred approach to accomplish this task. The proposed *lab* package aims to streamline the processing of laboratory

records, thereby enabling the generation of time series data suitable for subsequent analysis. To evaluate its efficacy, the package was tested using publicly available critical care data sourced from MIMIC-III (*Johnson et al., 2016*) and a multi-institutional medical care database known as the Chang Gung Research Database (*Tsai et al., 2017*; *Shao et al., 2019*; *Chiang, 2022*).

In the era of digital health, artificial intelligence has been successfully employed across various medical domains (*Shortliffe & Sepúlveda, 2018*; *Tseng et al., 2019, 2020*; *Rajkomar, Dean & Kohane, 2019*; *Emanuel & Wachter, 2019*). Preprocessing medical records is crucial in developing artificial intelligence–assistive services (*Harris et al., 2018*; *Sarwar et al., 2023*). Data correctness is a critical aspect of data quality that determines the suitability of EHR data for its research purposes (*Weiskopf & Weng, 2013*; *Verheij et al., 2018*; *Sarwar et al., 2023*). While data from EHR may be accurate at the source, choices made during the data extraction process can impact the results of research queries (*Denney et al., 2016*; *Agniel, Kohane & Weber, 2018*).

To improve the data processing step for EHR sharing or analysis, various guidelines and pipelines have been proposed for acquiring data from EHRs (*Rea et al., 2012*; *Knake et al., 2016*; *Maletzky et al., 2022*; *Miao et al., 2023*). Additionally, for processing and analyzing specific datasets like MIMIC-IV, multiple tools have been developed to cater to the unique characteristics of these datasets (*Mandyam et al., 2021*; *Gupta et al., 2022*). Furthermore, several R packages have been developed to assist in general EHR analyses, but they mainly focus on diagnosis or procedure data (*Springate et al., 2017*; *Wasey & Lang, 2022*), including the *dxpr* package (*Tseng, Chiu & Chen, 2021*). Another R package, *cleanEHR*, was specifically designed for creating a linkable database across multiple ICUs, although it does not include time-series data processing (*Harris et al., 2018*). Consequently, there is a need for a tool dedicated to time-series laboratory record analysis. In response to this gap, we developed the *lab* package to assist analysts in retrieving critical information from laboratory records and analyzing the differences in laboratory test results in different research groups.

The proposed package has limitations. First, users are required to provide a mapping table to establish a link between their local laboratory codes and LOINC. Acquiring such mapping tables can be challenging, potentially hindering the ability to connect laboratory records with LOINC codes effectively. If LOINC mapping cannot be completed, users should provide reference ranges or other information when performing some functions related to the data provided by LOINC, such as reference ranges. However, it is essential to note that distinguishing between normal and abnormal laboratory test results may not always be necessary for laboratory record analyses. Second, we introduced an easy-to-use package that assists analysts in processing raw laboratory records. However, it is worth mentioning that missing values in medical measurements often exhibit informative patterns that can be leveraged for prediction purposes. Exploring the underlying dynamics of these patterns can be valuable in preparing datasets for subsequent machine learning models (*Li et al., 2021*). Furthermore, the optimal imputation approach may vary depending on the specific dataset being analyzed (*Hu et al., 2017*). Although we have implemented common imputation approaches used in clinical research (*Jakobsen et al.,*

*2017*; *Hu et al., 2017*), users should consider alternative imputation algorithms if there are any more suitable ones for their particular datasets. Nevertheless, even with the availability of this package, analysts must possess a comprehensive understanding of the data to determine the most appropriate method for data processing accurately.

## CONCLUSIONS

The proposed package assists clinical data analysts in overcoming the challenges of heterogeneous and sparse laboratory records. It simplifies laboratory record preprocessing and facilitates the generation of analysis-ready time series data. By utilizing this package and conducting additional analysis, researchers can understand disease progression in-depth, make informed treatment decisions, and enhance prognostic and diagnostic predictions.

## ACKNOWLEDGEMENTS

We thank Cheng-Yu Chiang from the Department of Information Management at National Central University for testing the *lab* package.

### Funding

This research was supported by the National Science and Technology Council of Taiwan (NSTC 111-2636-E-A49-014 and 111-2628-E-A49-026-MY3). The funders had no role in study design, data collection and analysis, decision to publish, or preparation of the manuscript.

### Grant Disclosures

The following grant information was disclosed by the authors:
National Science and Technology Council of Taiwan: NSTC 111-2636-E-A49-014 and 111-2628-E-A49-026-MY3.

### Competing Interests

The authors declare that they have no competing interests.

### Author Contributions

- Yi-Ju Tseng conceived and designed the experiments, performed the experiments, analyzed the data, performed the computation work, prepared figures and/or tables, authored or reviewed drafts of the article, and approved the final draft.
- Chun Ju Chen performed the experiments, performed the computation work, prepared figures and/or tables, authored or reviewed drafts of the article, and approved the final draft.
- Chia Wei Chang analyzed the data, authored or reviewed drafts of the article, and approved the final draft.

## Data Availability

Data and code are available at GitHub: https://github.com/DHLab-TSENG/lab.

The sample codes, codes used for the use case, and the development version of the package are available at NYCU Dataverse:

Yi-Ju Tseng, 2023, "Replication Data for: lab: An R package for generating analysis-ready data from laboratory records", https://doi.org/10.57770/C7MNRH, NYCU Dataverse, V2.

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
