# Peer review of "lab: an R package for generating analysis-ready data from laboratory records"

_PeerJ Computer Science, doi:10.7717/peerj-cs.1528_

## Round 0.1 · original submission · Major Revisions

The reviewers have substantial concerns about this manuscript. The authors should provide point-to-point responses to address all the concerns and provide a revised manuscript with the revised parts being marked in different color.

Reviewer 1 ·

Basic reporting

The article "lab: An R package for generating analysis-ready data from laboratory records" appears to be well-written, organized, and informative. However, here are some revision and improvement suggestions:

Experimental design

Explanation of methods: The paper mentions briefly the imputation methods utilized by the imputeTimeSeriesLab function, but it would be beneficial to provide more information on how these methods operate and when they should be employed.

Validity of the findings

1. The abstract is concise and effectively written. However, it would benefit from a more detailed explanation of the methods and outcomes.

2. The introduction is professionally written and informative. However, it could be enhanced by providing more context regarding the difficulties of analyzing laboratory records and the significance of preprocessing these records for time-series analysis.

3. Some sentences could benefit from clarity and readability enhancements. For instance, line 261 should read "A function for plotting line charts, plotTimeSeriesLab, is available" rather than "A function for plotting line charts, plotTimeSeriesLab, is available as a means of..."

4. The conclusion is effectively written and informative. However, it could be improved by providing additional information regarding the potential applications and impact of the "lab" package in clinical research and practice.

Additional comments

The references are appropriately cited and pertinent. However, the authors may wish to include more recent research on the analysis of laboratory records and the application of R packages for preprocessing and analysis.

Cite this review as

Reviewer 2 ·

Basic reporting

The authors of this paper have presented an open-source R package called "lab" that aims to streamline and expedite the process of laboratory report extraction. The package offers various features, such as mapping local laboratory codes to international standards, annotating abnormal results, slicing data into time-series windows, summarizing data using descriptive statistics, imputing missing values, and generating analysis-ready data. The authors suggest that this package has the potential to aid clinical data analysts in generating simple and reproducible scripts.

The figure provided in the manuscript needs improvement, particularly Figure 1. It is not clear and could benefit from clearer labeling and descriptions to aid reader comprehension.

Experimental design

In the method section, lab package has a function named mapLOINC to map the lab data to the LOINC. Which method or algorithm this function used to realize the map? And what should do when there is not a LOINC term to map? How to solve the non-map items?

Line 151 "LOINC can be used to annotate abnormal results". What kind of abnormal results can be detected, is it the abnormal data type, date or outlier detection? Detailed method needs to be described to understand the advantage and limitations of the function.

Is it possible to integrate the abnormal detection function in the pipeline whenever the record is finalized or updated to checked the integrity and guide the data confirmation? And how does version control work with the package?

For data imputation, line 281 "if an individual never underwent a certain test, the mean of the test results of all individuals in the data set can be imputed." Use the metadata matched patients instead all patients for individual imputation would make more sense. I wonder if the author tested this and other implemented imputation methods on real-world lab records with mask.

For plotWindowProportion to visualize missing value and select optimal time window, is there a function to show the distribution of missing values with different time window? Then user can select the optimal time window with high efficiency.

Validity of the findings

The manuscript would benefit from more discussion on the potential impact of "lab" in clinical practice. Specific use cases where "lab" could be particularly useful for clinicians or researchers working with laboratory data should be discussed.

While the authors briefly mention other packages and approaches for laboratory report extraction, more detail is needed on how "lab" compares to these options. A more thorough comparison would help readers understand the strengths and limitations of "lab" and its place in the existing literature.

Additional comments

Attention should be paid to language and grammar issues throughout the manuscript to ensure clarity and comprehension. Awkward phrasing or grammatical errors can make comprehension difficult and may detract from the overall impact of the manuscript.

Cite this review as

Reviewer 3 ·

Basic reporting

The paper introduces an open-source R package called "lab" that enables analysts to retrieve detailed information from laboratory records, annotate abnormal results, slice data into time-series windows, summarize data using descriptive statistics, impute missing values, and generate analysis-ready data. The proposed lab package simplifies and accelerates the workflow of laboratory report extraction and may assist clinical data analysts in generating simple and clean scripts that can easily be shared and reproduced.

In term of clear and unambiguous, professional English used throughout the paper, the paper is mostly well written, easy to understand. However, there are a few areas to improve:

1. In the conclusion section "....can assist clinical data analysts in generating simple and clean scripts that can easily be shared and reproduced." I think "simple" can be removed from the sentence. In addition, I don't think it is necessary to emphasize that the generated scripts can be shared and reproduced, which are common features of open source softwares.

2. "Unlike diagnosis or procedure records, laboratory records contain information on the tests ordered and the results of the tests, which renders these records more difficult to process in EHR analyses" why test orders and results of the. tests are more difficult to process? Readers might not have context here. Please provide more context here.

3. In line 70, "the other" is not an appropriate usage here.

In terms of second criterial "Literature references, sufficient field background/context provided", the article provides sufficient background context and relevant literatures are properly referenced.

In terms of figures, tables and raw data, I noticed that some content of figure1 has has different font. The authors need to address this inconsistence.

The paper describes the conclusion clearly.

Experimental design

The paper developed an R package and applied it to two different datasets. there is no "experiment" involved.

Validity of the findings

The paper only stated the importance of having clean EHRs data but has no information about if there is any existing work to bridge the gap before. I believe the author can improve by providing more context about previous work. If there is no, the author can clearly emphasize the novelty of the work.

Regarding discussion and conclusion, In line 399-400, the author claimed that "The results indicate that this package can be used with laboratory records from different countries and institutions.", which is a very strong conclusion. I don't think the evidence provided by the author (using the R package against two datasets) can sufficiently support the claim. I notice that the author didn't include this sentence in the final conclusion section. in my opinion, the author can remove it from the main content as well.

Beside that, the conclusion section seems appropriate.

Additional comments

In conclusion, the paper is well written and easy to understand. There are minor issues with the language , which do not impact overall quality of the work. The conclusions are well stated with sufficient evidence. As I called out before, "The results indicate that this package can be used
400 with laboratory records from different countries and institutions." is a too strong claim based on the evidence. In my opinion, should be removed to be consistent with the conclusion section.

Overall, recommending to publish to minor revisions.

Cite this review as

---

## Round 0.2 · accepted · Accept

All concerns are addressed and I recommend accepting this manuscript.

Reviewer 1 ·

Basic reporting

The paper has addressed all problems from reviewer.

Experimental design

The paper has addressed all problems from reviewer.

Validity of the findings

The paper has addressed all problems from reviewer.

Cite this review as

Reviewer 2 ·

Basic reporting

The author responded to my comments and I have no further concern

Experimental design

The author responded to my comments and I have no further concern

Validity of the findings

The author responded to my comments and I have no further concern

Cite this review as

Reviewer 3 ·

Basic reporting

N/A

Experimental design

N/A

Validity of the findings

N/A

Additional comments

The revision mostly resolved my raised questions and concerns. Recommend to accept.

Cite this review as